# Contributions and Limitations of Biophysical Approaches to Study of the Interactions between Amphiphilic Molecules and the Plant Plasma Membrane

**DOI:** 10.3390/plants9050648

**Published:** 2020-05-20

**Authors:** Aurélien L. Furlan, Yoann Laurin, Camille Botcazon, Nely Rodríguez-Moraga, Sonia Rippa, Magali Deleu, Laurence Lins, Catherine Sarazin, Sébastien Buchoux

**Affiliations:** 1Laboratoire de Biophysique Moléculaire aux Interfaces, Gembloux Agro-Bio Tech, TERRA Research Center, Université de Liège, B5030 Gembloux, Belgium; afurlan@uliege.be (A.L.F.); yoann.laurin@uliege.be (Y.L.); magali.deleu@uliege.be (M.D.); l.lins@uliege.be (L.L.); 2Unité de Génie Enzymatique et Cellulaire, UMR 7025 CNRS/UPJV/UTC, Université de Picardie Jules Verne, 80039 Amiens, France; camille.botcazon@utc.fr (C.B.); nely.rodriguez.moraga@u-picardie.fr (N.R.-M.); catherine.sarazin@u-picardie.fr (C.S.); 3Unité de Génie Enzymatique et Cellulaire, UMR 7025 CNRS/UPJV/UTC, Université de Technologie de Compiègne, 60200 Compiègne, France; sonia.rippa@utc.fr

**Keywords:** plant plasma membrane, elicitor, lipid, amphiphiles, molecular interactions, biophysics, biomimetic membranes

## Abstract

Some amphiphilic molecules are able to interact with the lipid matrix of plant plasma membranes and trigger the immune response in plants. This original mode of perception is not yet fully understood and biophysical approaches could help to obtain molecular insights. In this review, we focus on such membrane-interacting molecules, and present biophysically grounded methods that are used and are particularly interesting in the investigation of this mode of perception. Rather than going into overly technical details, the aim of this review was to provide to readers with a plant biochemistry background a good overview of how biophysics can help to study molecular interactions between bioactive amphiphilic molecules and plant lipid membranes. In particular, we present the biomimetic membrane models typically used, solid-state nuclear magnetic resonance, molecular modeling, and fluorescence approaches, because they are especially suitable for this field of research. For each technique, we provide a brief description, a few case studies, and the inherent limitations, so non-specialists can gain a good grasp on how they could extend their toolbox and/or could apply new techniques to study amphiphilic bioactive compound and lipid interactions.

## 1. Introduction

Plants are fixed organisms, subject to many environmental constraints. In particular, they have to cope with a wide variety of pathogens. Unlike mammals, plants lack mobile cells dedicated to immune responses. They are protected by preformed physical barriers such as cuticular waxes on the plant scale, and cell walls on the cell scale. They also produce constitutive phytoanticipin compounds with antimicrobial properties [1]. Microorganisms that manage to bypass these defenses are then confronted with the innate immunity of plants, which can be stimulated by various types of molecules named elicitors. The plasma membrane (PM), separating the intracellular content from the outside, plays a central role in plants’ ability to detect microbes [2]. While many molecular patterns are known to be recognized by membrane receptors, some amphiphilic molecules directly interact with plant PM lipids while still triggering defense responses in plants [2]. Because they interact with the lipids from the plant PM, elucidating the mode of perception of these amphiphilic elicitors may require a specific approach compared to studying the receptor-recognized ones. In this review, we present an overview of several biophysical techniques especially well-suited to investigating the molecular interactions between amphiphiles and lipid membranes.

## 2. Specific Aspects of the Plant Plasma Membrane

The basic structure of PM, established from the fluid mosaic membrane model [3] common to all living organisms, is a lipid bilayer in which proteins are embedded or associated to via a variety of interactions, with a lipid-to-protein ratio of 1 to 1.4. Data accumulated since the publication of the fluid mosaic membrane model have revealed the unexpected and outstanding complexity of PM organization, and the essential role of lipids in the organization and intrinsic properties of PM, which appear to be crucial for ensuring its physiological functions. The great diversity of PM lipids [4] was revealed thanks to the development of lipidomics. Major classes of lipids are shared by all living organisms, such as glycerolipids (mainly phospholipids), sphingolipids, and sterols [4,5]. However, between species, cell types, or tissues within a species, the lipid composition of PMs can show a high degree of diversity, and plant PM exhibits further striking features. While animal PM essentially contains cholesterol, different phytosterols with diverse structures are present in plants [5]. The latter play significant roles in regulating the order level of the membrane. Concerning sphingolipids, sphingomyelin is absent in plants, and specific ceramides, named glycosyl-inositol-phosphoryl-ceramides (GIPCs), are the main plant sphingolipids, while totally absent in animal PMs. For example, in the model plant *Arabidopsis thaliana*, the plasma membrane is constituted of phosphatidylcholine (PC), phosphatidylethanolamine (PE), phosphatidylinositol (PI), phosphatidic acid (PA), phosphatidylserine (PS), digalactosyldiacylglycerol (DGDG), phophoinositides (PI) as glycerolipids, GIPCs with very long-chain fatty acids (up to 26 carbons), glucosyl ceramide and long-chain bases for the sphingolipid class, and sitosterol, campesterol, fucosterol, and stigmasterol together with conjugated sterols (sterylglucoside and acyl sterylglucoside) for the sterol class [5,6]. In plants, the heterogeneity of the spatial distribution of lipids and proteins at the PM surface has been established together with the presence of nano- to micro-scale domains exhibiting different order levels [7,8,9], and the differential ability of plant lipids to generate such a biophysical heterogeneity on model membranes was described [5]. The spatial segregation of proteins and lipids in resting state and their dynamic relocalization within PM nanodomains to promote functional signaling platforms, concomitant with modifications of PM order and fluidity, have been evidenced in immune signaling, host–pathogen interactions, and particularly documented in plant–microorganism interactions [7,8,9,10] (for a recent review, see Jaillais and Ott, 2020 [10]). Furthermore, the asymmetry of the lipid distribution between the two leaflets of animal and plant PM is another key feature of membrane organization and function. In animals, most of the available data on this asymmetry comes from red blood cells and is still not yet fully elucidated. In plants, very few publications partially examine these crucial questions. Work performed on oat root PM indicated that phospholipids dominate the cytosolic leaflet followed by total sterols, whereas the reverse order applies to the apoplastic leaflet of the oat root PM [11]. Investigating the molecular basis of the electrostatic characteristics of plant endomembranes, Jaillais et al. evidenced that PA and PS sensors accumulate at the PM cytosolic leaflet in *A. thaliana* root epidermis, together with PI 4-phosphate (PI4P) [12]. Recent data suggested that GIPCs might be mainly located in the outer leaflet of tobacco PM [6], but no indication about the localization of the different molecular species of either free sterols or lipid-associated fatty acids is currently available in the literature.

## 3. Involvement of the Plant Plasma Membrane in Triggering the Immunity Signaling Process

Cell-surface protein receptors of the PM, called pattern recognition receptors, perceive chemical compounds informing plant cells of need to defend themselves [13]. When activated by their ligands, these receptors form complexes with co-receptor proteins to trigger immune responses. Biotic attacks are therefore recognized by molecular signatures coming from pathogens, more generally microbes, or from plant cells themselves. They are called pathogen-, microbe-, or danger-associated molecular patterns [14,15]. These patterns elicit the establishment of an inducible defense response (pattern-recognition-receptor-triggered immunity, PTI). It corresponds to a suite of downstream defense mechanisms including production of reactive oxygen species, influx of extracellular calcium, kinase activations, and a transcriptional reprogramming [16]. Extracellular molecular patterns inducing PTI are part of the invasion patterns that also include molecular signals produced by beneficial microbes and effectors produced by pathogens bypassing PTI [17].

In addition to membrane protein recognition, membrane lipid dynamics are also involved in invasion pattern perception. The FLS2 transmembrane kinase receptor of the peptide flg22 from bacterial flagellin is less mobile in presence of its ligand [18]. FLS2 is heterogeneously distributed in the membrane and forms transient clusters with co-receptors after flg22 recognition within nanodomains [19]. An increase of the PM order was also described after induction of the signaling cascade induced by the different elicitors such as flg22, cryptogein, and oligogalacturonides. Cryptogein is, moreover, able to induce an increase in membrane fluidity [20].

## 4. Interaction of Amphiphilic Elicitors with the Plant Plasma Membrane

Although an increasing number of elicitor–receptor couples have been identified, this type of perception is not the only possible one [2]. For some amphiphilic compounds, the perception could be linked to a direct interaction with the lipid part of the PM. The peptide alamethicin from the biocontrol fungus *Trichoderma viride*, well-known to form pores in biomimetic membranes, induces defense responses in *A. thaliana*. Defense-gene-triggering capability is correlated to the length of the peptide, showing a link between the pore-forming activity and the bioactivity of the compound [21]. In a same way, bacterial protein hairpins induce defenses in several plants (cell death hypersensitive response, defense gene activation, and resistance enhancement towards pathogens) and are known to interact with lipids and to form pores in membrane models under some experimental conditions [22]. Necrosis and ethylene-inducing peptide 1-like (NLP) proteins are described to bind to GIPCs [23]. Elicitins from oomycetes, with typical features of microbe-associated molecular patterns, are known to bind sterols and other membrane lipids [24].

Amphiphilic-lipid-based compounds are also proposed to be perceived by the lipid fraction of the plant PM. Surfactins, iturins, and fengycins, which are cyclic lipopeptides produced by *Bacillus subtilis*, activate plant defenses and are described to interact with membrane lipids [25,26,27,28,29]. It has been proposed that surfactin perception and triggering of plant defense mechanisms rely on a lipid-driven process rather than a direct sensing by a high-affinity protein receptor. Surfactins with longer acyl chain lengths show stronger interactions with membrane models and also display a higher plant-defense-triggering activity [27]. Rhamnolipids, glycolipids secreted by the bacteria *Pseudomonas aeruginosa*, trigger defense and protection in different plants and can also interact with membrane lipids [30]. It was notably shown that rhamnolipids can form supramolecular complexes with membrane phospholipids [31]. Interaction of rhamnolipids with biomimetic phosphatidylcholine membranes has also been extensively studied [32,33,34,35,36]. Recently, a combination of biological and biophysical approaches demonstrated that the interaction of synthetic glycolipids with biomimetic PM correlates with the plant biological response [37]. Biophysical studies and molecular modeling simulations showed that rhamnolipids fit into plant PM models but do not significantly affect lipid dynamics [38]. Amphiphilic phyto-oxylipins can also interact with plant biomimetic PM by modifying the lateral organization of domains in a lipid-dependent manner [39].

## 5. Biophysical Studies of Amphiphiles and Plant Plasma Membrane Interactions

### 5.1. Biomimetic Membrane Models

As already mentioned, the plant PM has a complex lipid architecture with the presence of a vast diversity of lipid species and the existence of lipid domains. This complexity arises from (i) the presence of proteins, (ii) the asymmetrical distribution of the lipids between the apoplastic and cytosolic leaflets, and (iii) their specific lateral organization and dynamics [40].

Studies on living cells [8] can be useful for characterizing the plasma membrane at a sub-micrometric scale, e.g., to get information on membrane dynamics and ordering [8,20], but cannot provide information at a molecular or atomic level (e.g., compound penetration/location into membrane, specific interactions with particular lipids, chemical determinants involved in these interactions, etc.) [41,42]. One strategy to obtain this kind of information is to use artificially made lipid membranes, or model membranes, even if they will never be an exhaustive representation of a real plasma membrane. With the models, the aim is to have a versatile system with an easily tweakable lipid composition to (i) mimic certain aspects of natural membranes and (ii) obtain complementary information using biophysical techniques unsuitable for studies on living cells. For instance, infrared (IR) spectroscopy and nuclear magnetic resonance (NMR) are powerful techniques for determining the precise location of an amphiphilic molecule in a lipid model membrane, but they can hardly be applied to living cells because of the complexity of the resulting spectra. Model membranes are also very suitable for a step-by-step approach to study the importance of specific lipid classes on amphiphilic molecule–lipid interactions. The preparation of artificial membrane model is relatively simple, even for more lipid complex compositions, and the only requirement is to know the composition of the biological target membrane (e.g., the plant PM). Because the lipid composition of a membrane is plant-, tissue-, or even organelle-specific, this information may be scarce but a few examples exist in the literature (see, for instance, References [43,44,45] for *A. thaliana*). Of course, the closer to the real membrane the lipid composition is, the more it will be biologically relevant, with the important caveat that increased complexity of the model will result in a more complex interpretation of the biophysical data (e.g., Reference [38] for NMR). Hence, it is important to choose the lipid composition with an adequate trade-off between the membrane complexity (and thus biological relevance) and the interpretability of experimental data. It is noteworthy to mention that this is a limitation of the biophysical techniques rather than of the artificial membrane models.

Figure 1 presents some of the classical models that are used for the analysis of amphiphile/membrane interactions. We focus here on three kinds of artificial membranes that have been used to study the interactions between lipids and amphiphilic molecules: (i) liposomes, (ii) oriented bilayers, and (iii) lipid monolayers. Other models exist, like bicelles or supported lipid bilayers, but, as they are rarely used with the techniques presented in this review, they are not discussed. For a more complete description of the different classes of membrane models, the reader can refer to other reviews [46,47].

Liposomes are one of the most common models used to study membrane dynamics, phase behavior, membrane fusion, membrane permeability and integrity, and its interaction with exogenous molecules. Depending on their tridimensional structure, many classes of lipids, such as long-chain phosphatidylcholines, tend to form liposomes through self-assembling in an aqueous medium [48]. Depending on the protocol used and according to their size, four classes of liposomes are commonly employed in biophysics and are depicted in Figure 1A. Small unilamellar vesicles (SUVs) constitute the smallest liposomes, with a typical size ranging from 20 to 80 nm. Large unilamellar vesicles (LUVs) are bigger, with a diameter of 100 nm up to 1 µm. It is important to note the membrane lipids rarely self-aggregate to form SUVs or LUVs. Instead, their spontaneous aggregation leads to micrometer-scaled multilamellar vesicles (MLVs), which are thus easy to prepare. Upon extrusion though pored membranes, MLVs can be converted to LUVs or SUVs, depending on the size of the pores. Additionally, sonication or freeze/thaw cycles can be used to form SUVs from MLVs [49]. Finally, it is also possible to form giant unilamellar vesicles (GUVs), which share the same size as MLVs but have only one lipid bilayer. Different methods can be used to prepare GUVs, like electroformation, natural swelling, or gentle hydration (for more details concerning GUV preparation, see References [50,51]). Whether they are SUVs, LUVs, GUVs, or MLVs, liposomes remain relatively easy to form, even with complex lipid compositions, which constitutes a major advantage for biophysical studies. Even for some lipids such as sterols or some phosphatidylethanolamines that do not form liposomes on their own, it is still possible, for instance, to insert them into phosphatidylcholine membranes.

Oriented bilayers are models used mainly in solid-state nuclear magnetic resonance spectroscopy (SS-NMR) to determine the orientation and structure of peptides and proteins in a lipid environment [52]. To prepare oriented bilayers, lipids and peptides are dissolved in an organic solvent and sprayed onto stacked ultra-thin cover glasses. After removing the solvent, the samples are hydrated to form planar phospholipid membranes on glass slides [53] (Figure 1B). The main advantage is the unique orientation of the sample, perpendicular to the NMR field, which leads to a simplification of the resulting NMR spectra [54].

As suggested by their name, lipid monolayers are constituted of a single layer located at the air–water interface, and mimic the outer leaflet of the membrane. The hydrophobic hydrocarbon chains orient towards the air phase and are perpendicular to the interface, whereas polar head groups are immersed into the aqueous medium [46] (Figure 1C). To study their interactions with lipids, exogenous molecules are injected into the aqueous phase and diffuse freely in the system. Information concerning the adsorption kinetics, insertion and penetration of the compound, and the lipid’s capacity to attract it can be evaluated using this model. The main advantage of the monolayer model is the possibility to study a single lipid at a time in order to obtain valuable insight about lipid specificity of a given interaction. It is also the only method allowing studies with pure sterol not mixed with another lipid. Lipid monolayers are classically used in atomic force microscopy (AFM), Brewster angle microscopy, and tensiometry experiments, but are rarely employed in fluorescence and not at all in solid-state NMR spectroscopy.

Finally, one point to consider when trying to obtain more relevant artificial systems is the membrane asymmetry between the outer and inner leaflets. Asymmetrical membrane models have been developed in the last fifteen years, notably driven by the London [55,56] and Heerklotz [57,58] groups. They can be prepared by several methods, such as by using cyclodextrins as lipid carrier molecules [59], using enzymes [57], by different microfluidic technologies [60], or by hemifusion between models with different lipid compositions [61]. The development of these asymmetrical models has been mainly focused on mammalian lipid membranes. To our knowledge, there are no reports of using these models to mimic plant PM. This may be due to the lack of knowledge about the asymmetry of the lipid distribution in this system [40]. However, provided the lipid composition of each leaflet is characterized, virtually nothing prevents the use of asymmetrical membrane models in the context of the plant PM.

### 5.2. Solid-State NMR Spectroscopy

Based on the observation of nuclear spin behaviors in a magnetic field, SS-NMR is a powerful technique to characterize the behavior of biomolecules in a lipid environment [62,63]. This non-invasive and non-destructive tool allows information to be obtained about a broad range of parameters like biomolecule insertion and location inside the membrane [53,64,65] as well as their effects on lipid dynamics and membrane integrity [29,38,66,67]. The main drawback of SS-NMR is that it usually relies on isotopes with a substantially low natural abundance (^2^H, ^13^C, or ^15^N). As a consequence, artificially labeled molecules are quite often used, making it inherently more difficult to use with complex lipid compositions. Thus, the vast majority of the model membranes used in SS-NMR studies are composed of very few lipid species (one to three). Since SS-NMR, as any biophysical technique, provides information about molecular interactions, it can be applied to virtually any biological context. Here, we present some examples where SS-NMR was used to study molecules interacting with lipid membranes and that have or can be applied to amphiphilic elicitors [38].

#### 5.2.1. Structural Information

SS-NMR is quite useful for characterizing the structure and orientation of proteins and peptides in a lipid environment [68,69]. Compared to crystallography, SS-NMR has three great advantages: (i) sample preparation is much easier, without the fastidious crystallization step, (ii) it is compatible with more representative membrane models like liposomes [64,70] or oriented bilayers [65,71], and (iii) as hydration state can be maintained, it allows better biomimicry. SS-NMR is particularly adapted to characterizing the insertion and structures of peptidic elicitors in membrane models. ^15^N SS-NMR can be used to provide information about the orientation of this helical peptide with respect to the membrane surface just by looking at the location of the peaks in the NMR spectrum. For instance, in the case of ^15^N-labeled alamethicin in interaction with oriented bilayers, the ^15^N NMR spectrum indicated that the peptide is oriented perpendicularly to the membrane surface, which was interpreted as its insertion inside the lipid bilayer [53,65].

Additional information such as the tilt angle (between the peptide long axis and the normal to the plane of the membrane), the azimuthal angle (rotation angle around the peptide long axis), and the peptide secondary structure can be determined using a two-dimensional ^1^H-^15^N NMR experiment called PISEMA (polarization inversion and spin exchange at the magic angle) [72,73,74]. Using this experiment, Salnikov and coworkers showed that alamethicin adopts mixed α-/3_10_-helical structures into palmitoyl-oleyl-phosphatidyl-choline (POPC) bilayers [65].

#### 5.2.2. Information on Lipid Dynamics

SS-NMR spectroscopy is one of the most suitable techniques to study lipid dynamics in membrane models. Data on the polar head group are obtained using ^31^P NMR (in natural abundance), while ^2^H NMR on deuterated lipids gives insights into the dynamics of the hydrophobic core [75,76]. In both cases, the information is easily extracted from the spectral width and shape, which are dominated by the chemical shift anisotropy (CSA, Δσ) in ^31^P NMR and by the quadrupolar splitting (Δν_Q_) in ^2^H NMR (Figure 2A). Qualitatively, and for both ^31^P and ^2^H NMR, the interpretation of the spectrum is quite straightforward, as the spectral width and the lipid dynamics are inversely proportional. Thus, an increase of lipid dynamics (or a decrease of the order) leads to a decrease of the spectral width and vice versa. This is illustrated Figure 2B.

In addition to the qualitative interpretation, quantitative information can also be extracted from SS-NMR spectra of lipid membranes. Undeniably, the most used quantitative parameter is the order parameter S_CD_ (C and D standing for carbon and deuterium, respectively), as it transcribes the fluctuation of the orientation of the C–^2^H bond. Numerical values for S_CD_ can range from 0 (highly mobile C–^2^H bond) to 1 (no mobility) [77]. Hence, a decrease of S_CD_ values due to an exogenous elicitor means an increase of lipid disorder and dynamics, whereas a rise of S_CD_ reflects an increase of the lipid acyl chain rigidity. Experimentally, individual S_CD_ values (one value per deuterated carbon position) are easily extracted from each quadrupolar splitting, Δν_Q_, visible in a ^2^H NMR spectrum. By plotting the S_CD_ values against the carbon position, one can visualize the order profile for the lipid membrane and thus gain access to the precise dynamics along the lipid chains, as depicted on Figure 3.

When direct extraction of the order parameter is not possible (e.g., the spectrum is too noisy) or not necessary, one can still gather the average lipid dynamics thanks to the spectral moments [76] and, more specifically, the first moment, M_1_. M_1_ is calculated directly from the NMR spectrum, and is proportional to the membrane-averaged quadrupolar splitting <Δν_Q_>. As for the Δν_Q_ (or S_CD_), a high value of M_1_ is characteristic of a rigid membrane where the lipid dynamics are rather low. M_1_ is thus particularly useful to quantify the average dynamic state of a lipid membrane. Likewise, plotting M_1_ against the temperature is useful for analyzing the changes in the lipid dynamics along with the temperature, and for determining the phase transition temperature, T_m_ , where the lipid chains undergo a transition from almost static (gel phase) to highly mobile and disordered (fluid phase) (Figure 4). Such a transition seldom occurs in vivo, where biological functions require a well-balanced amount of lipid mobility. Because a molecule that alters T_m_ has a direct impact on the lipid dynamics at a given temperature, it may enhance or reduce any biological functions that depend on it (e.g., signal transduction).

Using ^2^H SS-NMR and such M_1_ analysis, Monnier and coworkers noted a sterol-dependent fluidization of a plant PM model induced by rhamnolipids [38]. Indeed, a decrease of the spectral width was observed in the case of an addition of rhamnolipids to a model containing stigmasterol, whereas no variation of ^2^H NMR spectrum shape was observed for the same experiment using a model with β-sitosterol. Likewise, by substituting phytosterols by the fungal ergosterol, a stronger increase in the lipid dynamics was noticed for the two plant PM models. These results highlight the impact of sterol nature on the membrane destabilization induced by rhamnolipids. Hence, by giving information at the molecular scale, ^2^H SS-NMR spectroscopy provides useful tools to better understand the biological activities of rhamnolipids, like their antifungal activity and their ability to trigger plant defenses. ^1^H MAS-NMR (magic angle spinning NMR) spectroscopy can also be helpful in studying the impact of elicitors on the temperature of the gel-to-fluid phase transition. By using this complementary approach, which does not necessitate labeled molecules, it was shown that two elicitors, alamethicin and mycosubtilin, lower the gel-to-fluid transition temperature for dimyristoyl-phosphatidyl-choline (DMPC) liposomes [26,78].

Membrane integrity can also be easily assessed by SS-NMR. Indeed, while liposomes give a broad ^31^P NMR spectrum, small vesicles or micelles (i.e., fast-tumbling objects) exhibit an identifiable narrow peak. This can be interesting when studying the destabilizing effect of some elicitors, like surfactin [28,29,67,79]. For example, Buchoux and coworkers used ^31^P NMR to study surfactin-induced destabilization of negatively charged DMPC/dimyristoyl-phosphatidyl-glycerol (DMPG) liposomes with a surfactin-to-lipid ratio as low as 0.02 [29]. This ratio is 10 times less than the one found by isothermal titration calorimetry and ^31^P NMR experiments on neutral POPC liposomes (solubilization to micellar structures is detected at a ratio of 0.22 using isothermal titration calorimetry and is characterized by the emergence of an isotropic peak on the ^31^P NMR spectrum) [67]. This difference highlights the critical influence of the lipid model when studying membrane-interacting molecules like surfactin.

Finally, one limitation of SS-NMR spectroscopy is the high complexity of the spectra signal when more than three to four classes of lipids are present. This is particularly true for lipid dynamics analysis and structural characterization of peptides in membranes. In ^2^H NMR, a loss of resolution was observed for a plant model containing six different lipid classes [38], which made the spectra significantly harder to interpret, even if only one lipid species was deuterated (and thus observed). For a detailed analysis of more complex systems, a combined approach with other biophysical techniques like molecular dynamics simulations and fluorescence can be considered in order to overcome their mutual limitations. Fluorescence spectroscopy and imaging are also interesting options to overcome a second limitation of SS-NMR, especially concerning the study of lipid systems with a coexistence of phases. Indeed, SS-NMR gives a global information on systems (e.g., concerning lipid dynamics, order, or type of phases). Coexistence of phases can be visualized in ^2^H NMR, but it is quite impossible or extremely difficult to quantify the enhancement or decrease of a specific phase due to the elicitor action by this technique. This specific local information is more accessible via fluorescence or molecular modeling approaches.

### 5.3. Molecular Modeling

Molecular modeling methods are widely used for the investigation of biomolecule/membrane interactions at the atomic level. Several tools have been developed over the years; they mainly vary in the way molecules are represented and interact, and in the subsequent molecular information obtained. In this review, we focus on two particular methods that have been used to specifically study plant PMs and their interaction with bioactive molecules (see Figure 5 for graphical depiction). The first method, named docking, consists of the systematic analysis of the interaction of lipid molecules around a target of interest, thus mimicking a molecule inserted into a lipid monolayer. The second method, molecular dynamics (MD) simulations, can be used to investigate the dynamics of the molecule in a bilayer. Since molecular modeling can be used in many biological contexts, this section focused on modeling as a powerful toolbox with which to study molecules in membranes in order to give a good overview of what it can bring to the understanding the modes of action of amphiphilic elicitors.

#### 5.3.1. Molecular Docking

Different docking methods exist in the literature, but few are dedicated to the interaction between lipids and biomolecules. A method developed in the 80s called Hypermatrix [80] has proven to be effective and is based on the systematic calculation of the interaction energies between the molecules of interest (for example, plant lipids and elicitor molecules), taking the individual orientations of the molecules at the hydrophobic/hydrophilic interface into account, and an empirical force-field simulating the hydrophobic energy [81]. This docking method, illustrated in Figure 5A, is particularly useful to compare the specific interactions of the molecule of interest with different lipid types [41,80,82]. This approach was improved few years ago by increasing the number of interacting partners and the total number of molecules in the system; this variation is called the “big monolayer” method [28]. This procedure notably leads to a more accurate visualization of lipid domains in a monolayer and the potential effects due to their interactions with biomolecules [41].

Since both methods are static, their main drawback is the fact that the molecule conformations are “frozen” and then are not modified in terms of internal coordinates following their mutual interaction. Despite this flaw, the results obtained with Hypermatrix and big monolayer techniques are in good agreement with various experimentally measured parameters such as the interfacial area in a monolayer, the specificity of interaction in terms of lipid species, or the effects on lipid organization [41,83,84].

Both docking methods were previously used in order to investigate various molecules interacting with membranes, and notably plant PM. They are complementary to experimental biophysical approaches (notably those described in this review) and provide insight into the atomic/molecular specificity for the interaction of biomolecules with lipids. For instance, Lenarčič and coworkers showed that microbial cytolysin NLP interacts specifically with the GIPC, and that it plays a role in host specificity [23]. By docking, it was shown that GIPC conformation and organization are important for protein interaction. For cyclic lipopeptides such as surfactin docking analyses highlighted privileged lipid partners for insertion and destabilization of the membrane, mainly dipalmitoyl-phosphatidyl-choline (DPPC) located at the DPPC/dioleyil-phosphatidyl-choline (DOPC) domain boundaries [28,41,85]. The insertion into palmitoyl-linoleyl-phosphatidyl-choline (PLPC) and sitosterol monolayers for synthetic rhamnolipids (RL)Alk-RL and Ac-RL was also analyzed using docking approaches, showing different interaction patterns with PLPC, due mainly to a carboxyl group present in Alk-RL [37]. In a same way, sugar-based bola-amphiphiles have been shown to interact less with cholesterol than with POPC, using these modeling approaches combined with experimental assays [86]. In the same way, a green biosurfactant, hexadecylbetaine chloride, revealed a preferred interaction with sphingomyelin compared to POPC (mammalian lipid models) [87]. Modeling has also been proven to be an efficient tool with which to elucidate the organization of small peptides in the membrane. Those approaches have also allowed molecules to be designed with specific membrane-interacting properties [88,89]. In the case of plant PM, only a few studies are available.

#### 5.3.2. Molecular Dynamics Simulations

MD is a much more computationally complex method based on Newton’s equations of motion. It gives details on the interactions at the atomic resolution, but also sheds light on the energetic and dynamic components of the process. It involves the use of a force-field to simulate the movements of atoms relative to each other. Force-fields are a collection of potential equations and various parameters to reproduce stretching, bending, and rotations of bonds as well as non-bonded interactions, such as electrostatics and Van der Waals. A wide range of force-fields are available, depending on the molecule type to be simulated [90,91,92]; they are integrated within various MD packages, such as GROMACS, AMBER, NAMD, or CHARMM [93,94,95,96,97]. Classical MD simulations with an all-atom representation have a typical duration or around 100 ns up to 1 µs for membrane size from around 100 up to 1000 lipids. Coarse-grained representations, in which small groups of atoms (three to four heavy atoms) are described using one bead per group, allowing reduced simulation time, improve sampling [98], and represent the membrane with up to several thousand lipid molecules from different species and for simulation times up to hundreds of microseconds. These two techniques have been extensively applied in the past decade to study mammalian, bacterial, and organelle membranes, leading to accurate representations of the PM, lipid nanodomain formation mechanisms, membrane dynamics (flip-flop for example), and perturbations induced by a wide range of active molecules (e.g., realistic membrane [99], rafts [100,101], flip-flops induced by a protein [102], peptide-induced curvature [103]). A recent review by Marrink [104] provided a great overview of the possibilities offered by MD simulations for lipids and membrane investigations. Specific parameters can be extracted from these simulations for comparison with experimental data such as area (or volume) per lipid, order parameters of acyl chains, lateral diffusion coefficients, electrostatic potentials, depth of membrane insertion, etc. These parameters help to confirm models and unveil new mechanisms of interaction.

In the case of plant PM, the integration and parametrization of specific plant lipids, such as GIPCs, into force-fields are still ongoing in order to provide more realistic plant-specific membranes. Despite the missing lipid topologies, simple plant membrane models (consisting of up to three to four different lipid species at most) in interaction with biomolecules have been obtained using MD simulations. A peptide from the Rem1.3 protein, involved in the protection of the plant against viral infection, has been shown by different approaches, including MD simulations, to have a preferential interaction with phosphoinositides from the plant PM, and this interaction is involved in lipid domain formation [105]. A very recent study showed that molecules called hydroperoxides, produced by plants under stress, are able to interact with a model membrane composed of PLPC, sitosterol, and plant glucosylceramide. Hydroperoxydes perturb the lateral organization of the membrane, and glucosylceramide is the privileged partner for lipid interaction [106]. Another study on rhamnolipids suggested that they can insert into a POPC/PLPC bilayer in a very specific manner, in accordance with experimental data [38]. In the light of MD simulations carried out on mammalian or bacterial membranes, which include a vast number of lipid species, simulating domain formation, sterol flip-flop (for animal membrane), lipid asymmetry, or other properties, it is clear that a realistic plant PM model is now the primordial next step required in order to reach a molecular understanding of its specificities regarding lipid dynamics, asymmetry, and interactions with bioactive molecules for comparison with other model membranes.

### 5.4. Fluorescence Spectroscopy and Imaging

Fluorescence spectroscopy is a classical technique used in biophysics to study the interaction between a biomolecule and a lipid vesicle. A fluorescent molecule called a fluorochrome is submitted to a radiation at a specific wavelength emitted by a laser. This radiation is absorbed by the probe and induces an electronic transition from ground state to an excitation state. Fluorescence occurs when the excited electron relaxes to its ground state by emitting a photon at a specific wavelength longer than the excited one. Excitation and emission wavelength are specific to each molecule (Table 1), which constitutes the main advantage of fluorescence spectroscopy. Indeed, the molecule of interest can be selectively excited, allowing studies on more complex systems such as living cells. In general, biological molecules and compounds used in membrane models are poorly fluorescent. To overcome this limitation, external fluorescent probes are classically used (Table 1). However, close attention must be paid to the amount of probe inserted. Some fluorochromes have a large and planar aromatic moiety and can disturb membrane dynamics if too concentrated. This is particularly true for fluorescent synthetic lipids where the probe is grafted onto the polar heads or the acyl chains. In general, the percentage of fluorescent probe does not exceed a few molar percent (typically 1–5 probes for 100 molecules). Another method used to study biomolecule–lipid interactions by fluorescence is to synthesize an analogue of the target compound containing a fluorophore part [107,108]. For example, the fluorescent cyanophenylalanine was grafted onto alamethicin to study its interaction with the membrane [107]. Likewise, another strategy can be to use the fluorescent properties of a complex of two molecules that do not emit fluorescence separately. This strategy was notably developed by Rausch and Wimley to study the leakage property of alamethicin on POPC vesicles. For this, they used lanthanide metal terbium(III) (Tb^3+^) and dipicolinic acid, which present a strong greenish emission when they are complexed [109]. Finally, it should be noted that certain elicitors possess an intrinsic fluorescence which can be directly used to characterize their interaction with lipids. As an example, fengycin is fluorescent thanks to the presence of tyrosine residues [110].

In contrast to SS-NMR, some fluorescence assays can be easily carried on plant cells or on models reconstituted from lipid extracts, given their selective excitation properties (previously mentioned). This selectivity allowed the development of fluorescence imaging techniques able to visualize the fluorescent dyes directly on cells. Information on lipid domains [111,112] and membrane organization (e.g., the coexistence of gel/fluid phases) [8] became available without the use of membrane models. In this sense, fluorescence is an ideal technique with which to obtain more local information about membrane organization. This section presents some approaches used for the study of elicitor–lipid interactions applied to plant cells or artificial membranes. We focus mainly on the application of fluorescence spectroscopy and imaging to analyze membrane permeabilization, organization, and dynamics. For a more complete review regarding fluorescence applications for studies of biological membranes in general, readers can refer to Reference [113].

#### 5.4.1. Membrane Permeabilization

Membrane permeabilization and leakage can be easily studied using calcein and carboxyfluorescein release experiments. With these methods, the elicitor is added to preformed liposomes that contain the fluorochrome in their aqueous compartment. Initially, the probe is self-quenched due to its high concentration within the liposomes and no fluorescence signal is observed. If the elicitor has an effect on the membrane model (e.g., membrane destabilization, pore formation, lysis), the probe leaks from the liposome and is thus diluted in the external medium, leading to an increase of emitting signal (Figure 6A). However, calcein and carboxyfluorescein release assays do not allow the precise mechanism of lipid perturbation to be assessed. Indeed, the enhancement of fluorescence intensity is non-linearly related to the amount of dye released and to the extent of the leakage [67]. As a consequence, other approaches must be applied to understand the molecular mechanism behind the fluorochrome release. The interaction between lipids and amphiphilic elicitors such as surfactin [114,115,118], rhamnolipids [35,119], alamethicin [120], fengycin [121], and the protein Harpin HrpZ [122] have been extensively studied using this method. For more details concerning mechanisms of liposome leakage induced by elicitors, readers can refer to studies published by Heerklotz and coworkers on the subject [67,114]. Some researchers have also highlighted the importance of lipid composition on liposome permeabilization induced by elicitors. For fengycin, Fiedler and Heerklotz [118] noted an inhibition of permeabilization for POPC liposomes containing PE or PG [118]. Similar experiments on surfactin showed that it promotes calcein release when PG is present, but inhibits the solubilization of liposomes that contain PE [118]. Similar results were observed by Uttlova and coworkers for the interaction of surfactin with liposomes containing different amounts of PG, PE, and PA. By studying surfactin influence on *Bacillus subtilis* lipid composition, a correlation with biological results was made by the authors, who noticed a decrease of PG amount in the presence of the elicitor, highlighting the adaptation of the bacteria [115]. As mentioned above, fluorescence techniques can be applied in models more representative of biological membranes. For example, Haapalainen and coworkers carried out calcein release experiments induced by Harpin HrpZ on vesicles prepared from *Arabidopsis thaliana* plasma membrane [122]. They observed a calcein leakage at an elicitor concentration ranging from 20 to 50 nM, which suggested the presence of pores formed by this protein.

#### 5.4.2. Information on Lipid Dynamics

Several techniques based on fluorescence spectroscopy are useful for the study of membrane dynamics in artificial membranes or living cells. For example, diphenylhexatriene (DPH) and trimethylammonium diphenylhexatriene (TMA-DPH) probes, respectively located close to the center of the bilayer and near the lipid/water interface [123,124], are classically used for steady-state polarization measurements. This method allows analysis of the lipid mobility and, as a consequence, the physical state of a lipid bilayer [125,126]. Indeed, the value obtained, directly related to the degree of freedom and mobility of the probe inside the membrane, differs according to the nature of the phase (Figure 6B). In the case of a gel state, high steady-state polarization values are obtained, due to the rigidity of the system. During the transition from gel to fluid phase, a large increase of rotational reorientation is observed, inducing an abrupt decrease of fluorescence polarization value [125]. Hence, plotting fluorescence polarization values as a function of temperature allows information to be obtained on the influence of an exogenous compound on the transition-phase temperature of the bilayer as well as on the global dynamics of each phases. For the latter, interpretation must be carried out carefully, especially in the case of a slight variation which could be due not to a change of lipid dynamics but to an interaction with the molecule of interest, which can modify the rotational motion of the probe. For example, Sanchez and coworkers observed a slight enhancement of probe polarization due to the incorporation of 10 mol% of di-rhamnolipids in DPPC vesicles [34]. Using Fourier-transform infrared (FTIR) spectroscopy, the authors suggested that this increase could be due to an interaction with the elicitors and not caused by an increase in membrane rigidity. Thus, fluorescence polarization measurement is not always a straightforward method and other approaches have to be used to have relevant interpretations. For gel-to-fluid transition measurements, significant differences of fluorescence polarization values are always observed between the two phases. As a consequence, elicitors’ influences on gel-to-fluid transition have been widely studied, notably concerning fengycins [116], alamethicins [125], and rhamnolipids [34,127].

Lipid mobility can be studied via steady-state fluorescence polarization experiments with probes directly grafted on the lipids. Depending on the fluorochrome used, the fluorescent part can be grafted onto the polar head or on the lipid acyl chains which allows lipid dynamics to be studied in different sections of the membrane (i.e., close to the water interface or deeper inside the hydrophobic core). For example, Kikukawa and Araiso used phosphatidylcholine-grafted 1,6-diphenyl-1,3,5-hexatriene (DPH-PC) and phosphatidylethanolamine-grafted nitrobenzoxadiazole (NBD-PE) (with the fluorescent moiety located, respectively, in the hydrophobic core and on the polar head) to measure the steady-state fluorescence polarization variation induced by alamethicin in interaction with POPC and DOPC vesicles [117].

Several probes, such as those from the boron-dipyrromethene (BODIPY) and DiANEPP families, can be used to visualize specifically ordered or disordered liquid phases [128]. They insert preferentially into ordered or disordered lipid phases, but do not allow the variation of lipid order to be quantified [128]. Lipid domains can also be visualized using Laurdan [129,130], which emits at a specific wavelength depending on the lipid phase [131]. Thus, Laurdan blues in ordered lipid phases and greens in disordered phases, with emission maxima at 440 and 490 nm, respectively [131,132]. Moreover, this probe is distributed equally between ordered and disordered phases and gives access to a mathematical parameter used to quantify membrane global order, GP_ex_, which is calculated according to the following equation.
(1)$${\mathrm{GP}}_{ex}=\frac{\left(I_{440}-I_{490}\right)}{\left(I_{440}+I_{490}\right)}$$

As presented previously for order parameters obtained from SS-NMR, GP_ex_ values increase with membrane order and enable the effects of an elicitor on the lipid order to be evaluated. When studying the variation of GP_ex_ at different excitation wavelengths ranging from 440 to 490 nm, the presence of a phase coexistence is reflected by an enhancement of GP_ex_ upon increasing excitation wavelength, whereas the presence of either a disordered or ordered lipid phase is indicated by a decrease and any modification of GPex values, respectively [28]. Deleu and coworkers studied the influence of surfactin on the dynamics of DOPC/DPPC (1/1), a model presenting a coexistence between gel and fluid phases at the temperature studied [28]. They observed different effects of surfactin on lipid models according to its concentration. For a concentration close to the critical micelle concentration (CMC), the elicitor inhibited the coexistence of phases and increasesd GP_ex_ value, suggesting an enhancement of order. At higher surfactin concentration, the effect on ordering decreased [28]. Di-4-ANEPPDHQ is another fluorescent probe sensitive to local lipid packing, which emits at 570 and 630 nm for ordered and disordered liquid phases, respectively [133]. As a consequence, the order of a lipid phase can be easily visualized by imaging, since the dye exhibits a green fluorescence in ordered domains and a red fluorescence in disordered domains ([132]; Figure 6C). Hence, membrane order level can be quantified using the “red-to-green ratio of the membrane”, which represents the ratio of emission fluorescence intensities recovered at 660 and 550 nm (I660/I550) [8,20,133]. Higher values have been observed for disordered liquid phases, whereas lower values have been noticed for the ordered liquid phase. Dinic et al. [134] noted that the presence of membrane proteins and peptides did not influence the spectra of Laurdan and di-4-ANEPPDHQ. As a consequence, these two fluorescent probes can be easily used in living cells. For example, Gerbeau-Pissot and coworkers used di-4-ANEPPDHQ to show that cryptogein induced an enhancement of PM order for tobacco BY-2 cells, but had no effect on *A. thaliana* PM [8,20]. It should be noted that a recent study showed that results obtained for GP_ex_ measurement can be skewed by the use of a di-4-ANEPPDHQ probe, due to its electrochromic properties (i.e., membrane potential dependence of the fluorescence emission spectrum) [135].

Fluorescence recovery after photobleaching (FRAP) is another classical fluorescence technique used to study membrane dynamics and to measure lipid lateral diffusion on artificial membranes or cells. A part of the membrane (or cell) is photobleached by a laser, which causes a total loss of fluorescence for probes located in this area (Figure 6D). Fluorescence recovery is subsequently measured to obtain information on the lateral mobility of probes. Fitting normalized fluorescence intensity as function of time allows the lateral diffusion coefficient for the fluorochrome in the membrane to be determined. As a consequence, information on fluidity, i.e., the measurement of rotational and translational motions within the membrane, becomes available. Using FRAP, Gerbeau-Pissot and coworkers observed an increase of PM fluidity for an addition of cryptogein in tobacco BY-2 cells [8]. Fluorescence correlation spectroscopy (FCS) is an interesting counterpart to FRAP, used to investigate membrane dynamics and lipid lateral diffusion. FCS measures fluctuations of fluorescence intensity in a defined volume, previously illuminated at a specific wavelength [131]. An autocorrelation function is obtained by correlating the signal (i.e., the intensity fluctuation) at the experimental onset time, t_0_, with the same signal after a lag time t_0_ + τ, (τ being the time interval) [131]. Hence, FCS can be used to investigate any process that leads to a change of fluorescence, such as a fluorochrome’s diffusion into and out of the detection volume. As an example, the influence of a molecule on gel and fluid phase dynamics can be easily measured using FCS [136], which makes it a good alternative to SS-NMR for gathering this kind of information. Compared to FRAP, FCS allows work at significantly lower fluorochrome concentrations (1 pM to 100 nM) thanks to its better sensitivity [131]. However, FCS signals are overly sensitive to the fluorochrome concentration and can deteriorate if this concentration is too significant. Using FCS is also more relevant than FRAP for the analysis of very fast motions (< µs), but is not well-suited to the study of slow-diffusing molecules [131,137]. Fluorescence cross-correlation spectroscopy (FCCS) is another method similar to FCS that can be used to measure the interactions between molecules that have fluorophores with different fluorescence emission wavelength [138]. In contrast to FCS, fluorescence intensity fluctuations of the two fluorochromes are detected separately using two different detectors. The auto-correlation function G(τ) correlates the fluctuation of the first probe at t0 with the fluctuation of the second after a lag time t0 + τ [131,138]. If G(τ) = 0, fluorescence intensity fluctuations of the two probes are different and they do not interact. On the other hand, if G(τ) ≠ 0, they are linked and diffuse together towards the detection volume. The amplitude of G(τ) depends on the fraction of the probes that are in interaction [138,139]. Hence, information on direct interaction and micro- and nano-domain formations can be obtained using FCCS [131]. Applying FCS-like techniques to plant cells remains challenging mainly due to the influence of concentration on FCS signals and background noise, two factors difficult to control in vivo [137]. However, in the past ten years, the scientific community has begun to develop alternative approaches based on FCS to overcome these problems. Biologists interested in the use of FCS-like measurement applied to plant cells can read the review written by Li and coworkers on the subject [137].

## 6. Conclusions

Biophysics provides tools that are perfectly tailored to the investigation of molecular interactions, especially in the case of lipid-membrane-bound amphiphiles. SS-NMR can provide a great deal of information about lipid dynamics or about the structure of a peptide amphiphile at the cost of having to rely on isotopically labeled molecules (e.g., ^2^H and ^15^N NMR), and can model membranes with simple lipid compositions. Fluorescence spectroscopy and imaging provide more general information on lipid dynamics (i.e., not at the lipid chain level) and also rely on (usually) non-natural fluorescence probes, but they can be used on much complex systems such as whole living cells. Molecular modeling describes molecules and lipid membranes with a finesse which simply cannot be achieved with any experimental approaches. However, it cannot really be used on its own and intrinsically relies on experimental validation and comparison. Combining these methods to counterbalance their respective limitations is thus particularly interesting when studying the modes of perception of amphiphilic elicitors.

Obviously, biophysical approaches cannot replace biological ones. In particular, due to their “bottom-up” design, they usually fail when the studied system becomes larger or more complex. Conversely, biology is good at studying more complex systems like cells, tissues, or even whole organisms, but its description of the interactions at the molecular level is weak at best. To assess the whole picture, it appears essential to apply a multidisciplinary approach combining biology, biochemistry, and biophysics, and we hope this review will help non-specialists to grasp how biophysical methods can be used in the context of amphiphilic elicitors.

## Figures and Tables

**Figure 1 plants-09-00648-f001:**
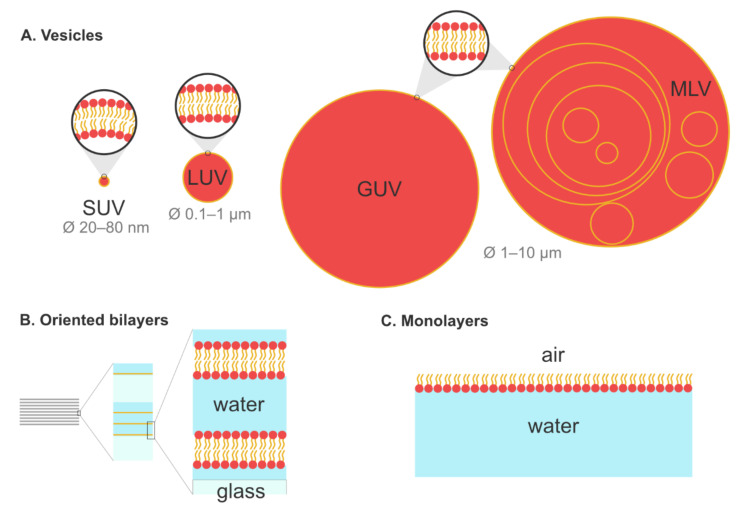
Graphical depiction of lipid self-assemblies classically used as membrane models in biophysical studies. (Panel **A**) shows the major types of liposomes and their typical diameter (ø) range: small unilamellar vesicle (SUV), large unilamellar vesicle (LUV), giant unilamellar vesicle (GUV), and multilamellar vesicle (MLV). Vesicular models are drawn to scale to illustrate their discrepancies in size and in terms of membrane curvature. (Panel **B**) shows oriented bilayers, where lipid bilayers are deposed on top of glass sheets separated by thin layers of water. (Panel **C**) shows how lipids can orient themselves at the air–water interface to form monolayers.

**Figure 2 plants-09-00648-f002:**
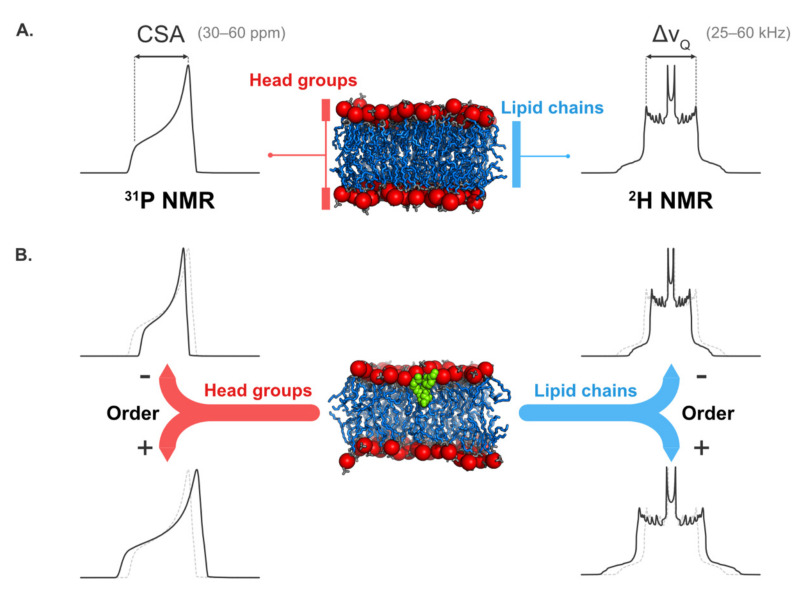
Information on membrane dynamics obtained using solid-state NMR spectroscopy. (Panel **A**) shows typical ^31^P NMR and ^2^H NMR spectra for a membrane. Thanks to chemical shift anisotropy (CSA), the ^31^P NMR spectrum gives details about the lipid head group dynamics. Similarly, the quadrupolar splitting (Δν_Q_) from a ^2^H NMR spectrum is linked to lipid chain dynamics. Typical values for both CSA and Δν_Q_ are given in parentheses. (Panel **B**) shows the effect that a molecule can have on membrane order when inserted into the lipid bilayer. For comparison and clarity, the NMR spectra when the molecule is absent are represented by the dotted gray lines. In both panels, the membrane is depicted using blue sticks for the lipid chains (gray for the rest of the atoms) and red beads for the phosphorus atoms. In panel B, the membrane active molecule is displayed using green beads.

**Figure 3 plants-09-00648-f003:**
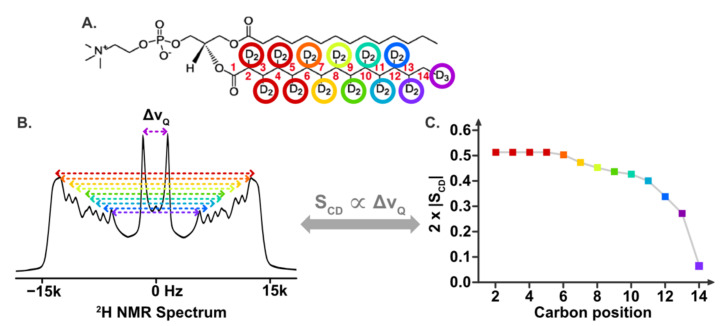
Illustration of how a ^2^H NMR spectrum can be used to obtain the order profile of a lipid membrane. (Panel **A**) shows a ^2^H-labeled molecule of dimyristoyl-phosphatidyl-choline, namely ^2^H_27_-DMPC, where all carbons from the sn2 chain are perdeuterated. To ease the interpretation of the figure by the reader, each deuterated methyl leading to a different quadrupolar splitting (Δν_Q_) is tagged by a different color. (Panel **B**) shows a ^2^H NMR spectrum where all the Δν_Q_ that are visible are labeled using the color that corresponds to the deuterated carbon position. (Panel **C**) shows the order profile where the S_CD_ value for each position is calculated directly from each Δν_Q_ value panel B, as S_CD_ and Δν_Q_ are proportional.

**Figure 4 plants-09-00648-f004:**
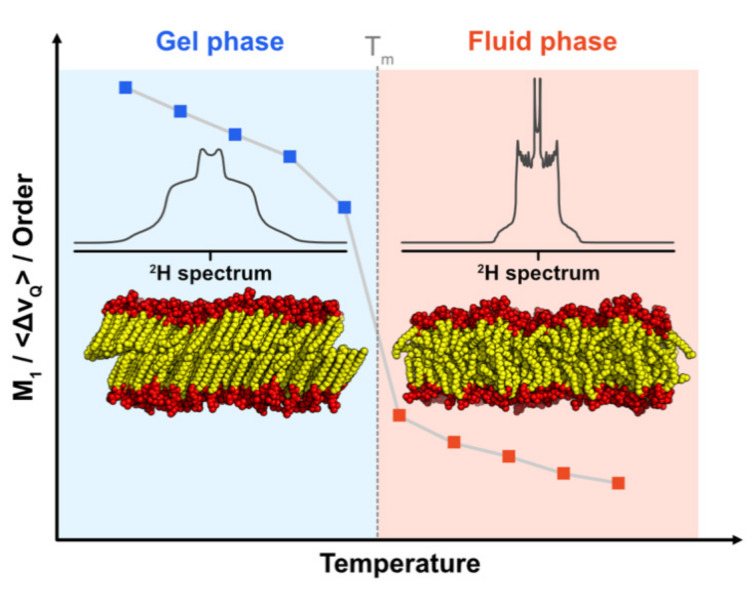
Influence of the temperature on lipid membrane dynamics. At low temperatures (**left** part), the lipids are in the gel phase where all the chains are fully elongated, leading to rather low dynamics (i.e., highly ordered), which is transcribed by ^2^H NMR in a characteristic wide spectrum. As a consequence, the values of M_1_ or <Δν_Q_>, derived from the spectral width, are also high. In contrast, at higher temperatures (**right** part), the lipids are in the fluid phase and the chains are quite disordered. The corresponding ^2^H NMR spectrum is narrower, leading to lower values for M_1_ and <Δν_Q_>. The temperature where the transition between the gel phase and the fluid phase occurs is noted as T_m_ (for “melting” temperature).

**Figure 5 plants-09-00648-f005:**
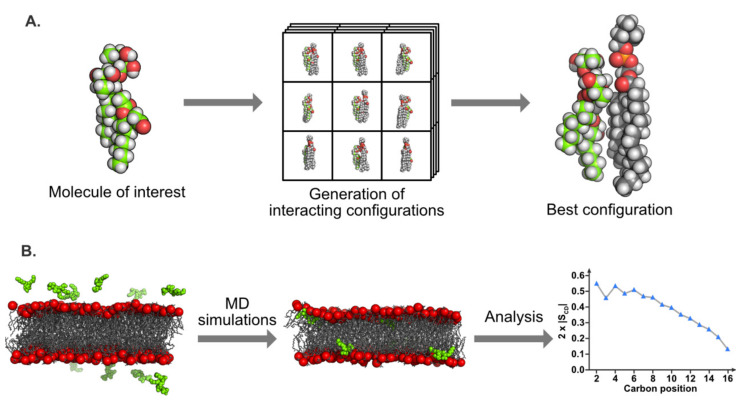
Examples of molecular modeling applied to amphiphilic molecule/membrane interactions. (Panel **A**) shows the Hypermatrix procedure where the molecule of interest is used as a reference for the docking of a lipid molecule (left). Many configurations of the reference molecule in interaction with the docked lipid are generated by translation and rotation of the lipid (middle). Among all these configurations, the one with the lowest energy (i.e., the most stable) is considered to be the best candidate with which to characterize the molecule/lipid interaction (right). In panel A, the carbons of the molecule of interest are represented using green beads, whereas lipid carbons are gray. For both molecules, hydrogen, oxygen, and phosphorus atoms are represented in white, red, and orange, respectively. (Panel **B**) shows a MD simulation of amphiphilic molecules (green beads) in interaction with a lipid membrane (chains as gray wires and head groups as red beads). At the beginning of the simulation (left), the molecules are located outside of the membrane (water is not represented for clarity). At the end of the simulation (middle), all the molecules are located inside the lipid bilayer. These qualitative results can be completed via quantitative analysis such the calculation of the order profile (right) that can then be compared with experimental data.

**Figure 6 plants-09-00648-f006:**
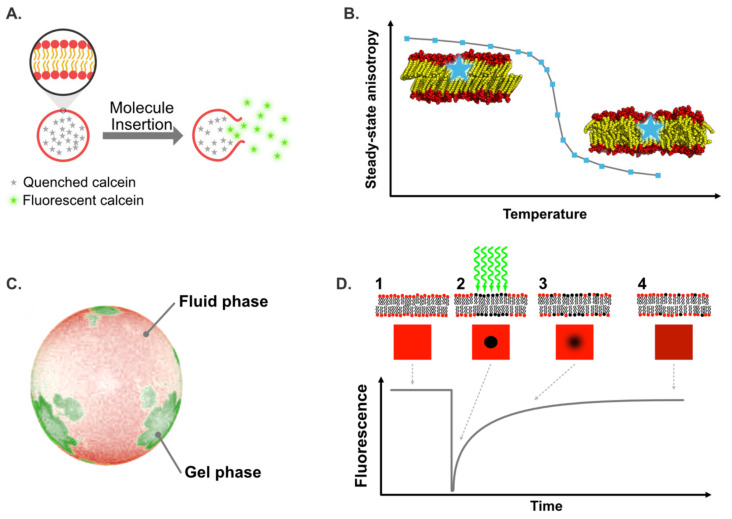
Different fluorescence approaches used for the study of bioactive molecule/membrane interactions. (Panel **A**) depicts a calcein release experiment where the membrane-active molecule is added to a solution of liposomes filled with a solution of calcein at a concentration high enough to be quenched (no fluorescence). If the added molecule destabilizes the lipid membrane, the calcein will be released outside of the liposome and its concentration will decrease enough that the probe will fluoresce. (Panel **B**) shows how the decrease of the steady-state anisotropy (see text for details) when the temperature rises can be used to characterize phase transition. (Panel **C**) shows DOPC/DPPC liposomes doped with two fluorescent probes: DOPE-Rho (red) and DPPE-NBD (green). As these probes are segregated to the fluid phase (for DOPE-Rho) and gel phase (for DPPE-NBD), they can be used to visualize and distinguish these phases. (Panel **D**) illustrates the different step of a fluorescence recovery after photobleaching (FRAP) experiment: (1) fluorescent lipid membrane at equilibrium (maximum and steady fluorescence), (2) fluorescent lipids are locally photobleached by a light pulse, (3) as the lipid diffusion occurs, the photobleached lipids and fluorescent lipids are mixed and the bleached area blurs out until (4) the fluorescence becomes uniform again.

**Table 1 plants-09-00648-t001:** Fluorescent probes commonly used to investigate membrane dynamics. Abbreviations: Di-4-ANEPPDHQ: AminoNaphthylEthenylPyridinium derivative; DPH: 1,6-diphenyl-1,3,5-hexatriene; DPH-PC: phosphatidylcholine-grafted 1,6-diphenyl-1,3,5-hexatriene; Laurdan: 6-dodecanoyl-N,N-dimethyl-2-naphthylamine; NBD-PE: phosphatidylethanolamine-grafted nitrobenzoxadiazole; TMA-DPH: 1-(4-trimethylammoniumphenyl)-6-phenyl-1,3,5-hexatriene *p*-toluenesulfonate.

Fluorescent Probes	λ_excitation_ (nm)	λ_emission_ (nm)	Membrane Location	Information	Refs
Calcein	495	515	Aqueous core	Permeabilization/solubilization	[67,114]
Carboxyfluorescein	490–500	515–520	Aqueous core	Permeabilization/solubilization	[115]
Di-4-ANEPPDHQ	488	560–570 (L_β′_ phase) 610–630 (L_α_ phase)	Membrane surface	Lipid order, lipid phases, membrane dynamics	[8,20]
DPH	358	430	Hydrophobic core	Membrane dynamics, gel-to-fluid transition temperature	[34,116]
DPH-PC (“lipid-like”)	350	430	Hydrophobic core	Lipid dynamics	[117]
Laurdan	340	440 (L_β′_ phase) 490 (L_α_ phase)	Membrane surface	Lipid order, lipid phases, membrane dynamics	[28]
NBD-PE (“lipid-like”)	450	560	Lipid/water interface	Lipid dynamics	[117]
TMA-DPH	360	435	Lipid/water interface	Membrane dynamics, gel-to-fluid transition temperature	[34]

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
