# Peer review of "Contributions and Limitations of Biophysical Approaches to Study of the Interactions between Amphiphilic Molecules and the Plant Plasma Membrane"

_plants, 2020, doi:10.3390/plants9050648_

Round 1
Reviewer 1 Report
The manuscript of a review article for MDPI Plants of Furlan et al. (Contributions and limitations of biophysical approaches to study plant plasma membrane perception of amphiphilic elicitors) brings a very refreshing overview in the field that is not easy and stays at the border between biophysics, biochemistry and biology. The review is perfectly focused and structured and it is very carefully written. The primary attention is to provide in an accessible form the overview on methods and tools that are available for testing the interplay between elicitors and lipidic biomembranes. I like the fact that the review is summarizing the whole panel of techniques that are used in the up to date research of membrane biology and biophysics. From SS-NMR, through modelling approaches to the methods utilizing fluorescence of marker molecules. I have only very minor comments and congratulate authors to such a valuable review.
1) I think that in Fig. 2 and Fig. 3 the lettering used in plot legends is too small.
2) Line 528 – The introduction of FRAP is kind of very brief, but frankly speaking, FRAP has limited “potential” in qualitative studies of processes connected with the lateral PM diffusion, so I understand why it is very brief. However, I would perhaps at least mentioning metho like FCS, FCCS and RICS, which collectively have a high impact in these studies.
3) Line 550 – First sentence has a missing full stop.
4) Reference 105 - letters should not be capitalized
Author Response
Summary and general remarks by reviewer #1:
The manuscript of a review article for MDPI Plants of Furlan et al. (Contributions and limitations of biophysical approaches to study plant plasma membrane perception of amphiphilic elicitors) brings a very refreshing overview in the field that is not easy and stays at the border between biophysics, biochemistry and biology. The review is perfectly focused and structured and it is very carefully written. The primary attention is to provide in an accessible form the overview on methods and tools that are available for testing the interplay between elicitors and lipidic biomembranes. I like the fact that the review is summarizing the whole panel of techniques that are used in the up to date research of membrane biology and biophysics. From SS-NMR, through modelling approaches to the methods utilizing fluorescence of marker molecules. I have only very minor comments and congratulate authors to such a valuable review.
Comment from the authors:
Thank you for your positive comments.
Specific remarks from reviewer #1 and response from the authors:
1) I think that in Fig. 2 and Fig. 3 the lettering used in plot legends is too small.
Response:
There were several issues with the figures that both reviewers pointed out. For that matter, all the figures have been updated. They now should be readable.
2) Line 528 – The introduction of FRAP is kind of very brief, but frankly speaking, FRAP has limited “potential” in qualitative studies of processes connected with the lateral PM diffusion, so I understand why it is very brief. However, I would perhaps at least mentioning metho like FCS, FCCS and RICS, which collectively have a high impact in these studies.
Response:
As advised and because FCS and FCCS methods can be more relevant than FRAP in some cases, we added a description for both of them.
Since none of the authors is familiar with RICS, we decided not to add it to avoid writing something wrongly interpreted.
3) Line 550 – First sentence has a missing full stop.
4) Reference 105 - letters should not be capitalized
Response:
Both these typos were corrected.
Reviewer 2 Report
The manuscript by Furlan et al. attempts to provide an overview of biophysical approaches that can be used to study the interactions between amphiphilic molecules and plant lipid membranes. Three types of experimental techniques are covered: solid-state NMR spectroscopy, molecular modeling (molecular docking and molecular dynamics simulations) and fluorescence spectroscopy and imaging. The authors also include a chapter on biomimetic membrane models; however, these are models, not an experimental technique (so this chapter should be numbered differently). For each of the methodologies, the authors reference some studies with plant or other elicitors, although the connection with plants is not always clear.
The manuscript is mainly focused on detailed explanations of different biophysical approaches; the explanations are for the most part clear enough for a non-expert reader. The figures need to be improved; see specific comments below. However, this review requires a better explanation of what the approaches that are presented can reveal specifically about plant membrane biology. This is mainly done by referencing different studies. I would highly recommend that the authors spend a bit more time on some of the more interesting studies that they reference, to explain with a specific case what exactly was learned by using a certain technique, and what are the caveats and limitations. The studies by Lenarčič, 2017 or Bücherl, 2017 are good examples. A better explanation of the composition and properties of the plant plasma membrane is needed.
Specific comments
1) I find some terms used in the review somewhat esoteric. In particular, I suggest changing the title and using the word ‘interaction’ instead of ‘perception’. ‘Eilicitors’ is also not a commonly used term; the authors should at least define at the beginning of their review what they mean by ‘amphiphilic elicitors’. My suggestion for the title would be “Contributions and limitations of biophysical approaches to study the interaction between amphiphilic molecules and the plant plasma membrane”.
2) Chapter 2, ‘Specific aspects of the plant plasma membrane’, has to be expanded and more thoroughly researched, so that it can be related with parameters studied by biophysical techniques. Composition, organization, and biophysical properties of the plant plasma membrane should be more thoroughly explained; and some comment on the differences between plants and different plant tissues would also be useful. The lack of description and detail here is problematic for the whole review. In line 269, for example, it is stated that plant PM is neutral; this is most certainly not the case. A good recent review that I can suggest would be Jaillais & Ott, Plant Physiol 2020, with references therein.
Later in the text, the authors often mention different phases and phase transitions; it is not explained at all how these pertain to the plant plasma membrane.
3) Maybe the term ‘perception’ can be used when describing an interaction with a receptor in the PM; since the authors discuss interactions with the lipid component of the PM, I would strongly recommend more technical terms ‘interaction’ or ‘binding’.
4) Chapter on biomimetic membrane models: the authors do not comment at all on the lipid composition of model membranes. Only in line 160-161 they mention model membranes made solely of sterols; I fail to see how such a model could be informative about the processes occurring at the plant PM.
This his highly relevant for the stated goal of this review, which discusses contributions and limitations of biophysical approaches.
5) In general, figure legends are insufficient, figures are sloppily prepared and labelled. The use of fonts and size is inconsistent even within the same figure, let alone between figures
Figure 1: Please add a more specific title and some description. What is SUV/LUV/GUV/MLV, what does Ø refer to? Are the structures drawn to scale in panel A? What is the range of sizes for GUVs? In line 137, µM should be changed to µm. Panels are labelled A,B and C while in text they are i, ii, and iii.
Figure 2: Can there also be a change in the polar head dynamics?
Figure 3: Please define DMPC. There are many other labels in the figure that are not explained or too small to read. ‘Depaking’ is probably ‘Depeaking’?
In legend you can say ΔνQ is directly proportional to Scd
Figure 4: Title? Label “A” should be moved. In the text, please comment on the relevance of these phase transitions for the plant PM.
Figure 5: Title: Molecular modelling approaches. Inconsistent labelling, use A, B and C.
Use the same molecular representation in panels A and B. This figure is particularly sloppy and uninformative. In panels A/B, I don’t understand what is docked on what. The lower panel doesn’t explain much about molecular dynamics simulations. The equations are not explained at all, so I am not sure how they will be useful to a reader. A specific example, describing what information was obtained from the simulation, would be much more useful. I suppose that what is shown is an actual simulation, so why not explain it?
Figure 6: More explanation is needed. Panel A is not clear; what is the molecule above the arrow? What are the black and yellow symbols representing? What is plotted in panel B? What are the probes shown in panel C, what is the difference between the left and the right image?
6) Abbreviations throughout the text: some are defined twice, some are used only once (therefore not needed), many are not defined. This will be particularly confusing for a non-expert reader.
Minor comments:
313 I don’t know what the authors mean by “membrane activity”
326 Period after “lipid” is missing
395 “a few molar percent” clarify range of concentration.
Some sentences were difficult to understand.
254 …to study the influence of elicitors on the temperature of the gel-to-fluid phase transition as it is sensible to lipid dynamics.
326 For instance, the study of the microbial cytolysin NLP has notably shown that plant specific glycolipids, GIPCs are involved in the recognition of the protein (should be referenced to Lenarcic 2017)
440 Similar experiments on surfactin showed an inhibition of solubilization for liposomes containing PE but an enhancement of calcein release in presence of PG [101].
489/490 Bluishes/greenishes are not English words
In general, the manuscript requires editing for English language and clarity.
Author Response
Summary and general remarks by reviewer #2:
The manuscript by Furlan et al. attempts to provide an overview of biophysical approaches that can be used to study the interactions between amphiphilic molecules and plant lipid membranes. Three types of experimental techniques are covered: solid-state NMR spectroscopy, molecular modeling (molecular docking and molecular dynamics simulations) and fluorescence spectroscopy and imaging. The authors also include a chapter on biomimetic membrane models; however, these are models, not an experimental technique (so this chapter should be numbered differently). For each of the methodologies, the authors reference some studies with plant or other elicitors, although the connection with plants is not always clear.
The manuscript is mainly focused on detailed explanations of different biophysical approaches; the explanations are for the most part clear enough for a non-expert reader. The figures need to be improved; see specific comments below. However, this review requires a better explanation of what the approaches that are presented can reveal specifically about plant membrane biology. This is mainly done by referencing different studies. I would highly recommend that the authors spend a bit more time on some of the more interesting studies that they reference, to explain with a specific case what exactly was learned by using a certain technique, and what are the caveats and limitations. The studies by Lenarčič, 2017 or Bücherl, 2017 are good examples. A better explanation of the composition and properties of the plant plasma membrane is needed.
Comment from the authors:
Thank you for your constructive remarks that helped us improve our manuscript.
We hope we managed to overcome the limitations and the issues you raised.
Specific remarks from reviewer #1 and response from the authors:
1) I find some terms used in the review somewhat esoteric. In particular, I suggest changing the title and using the word ‘interaction’ instead of ‘perception’. ‘Eilicitors’ is also not a commonly used term; the authors should at least define at the beginning of their review what they mean by ‘amphiphilic elicitors’. My suggestion for the title would be “Contributions and limitations of biophysical approaches to study the interaction between amphiphilic molecules and the plant plasma membrane”.
Response:
Using `perception` which implies a biological response was a mistake since biophysical studies (and thus our review) focus on molecular interactions.
We thank the reviewer for pointing that out and changed the title accordingly.
We also changed the term 'elicitor' and used 'amphiphilic molecules' instead and add a definition of elicitors within the first lines of the introduction.
2a) Chapter 2, ‘Specific aspects of the plant plasma membrane’, has to be expanded and more thoroughly researched, so that it can be related with parameters studied by biophysical techniques. Composition, organization, and biophysical properties of the plant plasma membrane should be more thoroughly explained; and some comment on the differences between plants and different plant tissues would also be useful. The lack of description and detail here is problematic for the whole review. In line 269, for example, it is stated that plant PM is neutral; this is most certainly not the case. A good recent review that I can suggest would be Jaillais & Ott, Plant Physiol 2020, with references therein.
Response:
This section has been expanded to present the lipid composition of the plant PM and how it differs from the animal PM.
Since our review is not about the lipid composition of the plant PM we tried to keep the description succinct enough so the link with the biophysical part is more obvious and while pointing some references (such as the really good and recent review from Jaillais and Ott) so the reader can find further information on this topic. Here is the addition to the manuscript:
Major classes of lipids are shared by all living organisms, such as glycerolipids (mainly phospholipids), sphingolipids and sterols [4,5]. However, between species, cell types, or tissues within a species, the lipid composition of PM can show a high degree of diversity and plant PM further exhibits striking features. While animal PM essentially contains cholesterol, different phytosterols with diverse structures are present in plants [5]. The latters play significant roles in regulating the order level of the membrane. Concerning sphingolipids, sphingomyelin is absent in plants and specific ceramides, named glycosyl-inositol-phosphoryl-ceramides (GIPCs), are the main plant sphingolipids while totally absent in animal PM. For example, in the model plant Arabidopsis thaliana, the plasma membrane is constituted of phosphatidylcholine (PC), phosphatidylethanolamine (PE), phosphatidylinositol (PI), phosphatidic acid (PA), phosphatidylserine (PS), digalactosyldiacylglycerol (DGDG), phophoinositides (PI) as glycerolipids, GIPCs with very long-chain fatty acids (up to 26 carbons), glucosyl ceramide and long-chain bases for the sphingolipid class, and sitosterol, campesterol, fucosterol, stigmasterol together with conjugated sterols (sterylglucoside and acyl sterylglucoside) for the sterol class [5,6]. In plants, the heterogeneity of the spatial distribution of lipids and proteins at the PM surface has been established together with the presence of nano- to micro-scale domains exhibiting different order levels [7–9] and the differential ability of plant lipids to generate such a biophysical heterogeneity on model membranes was described [5]. Spatial segregation of proteins and lipids at resting state and their dynamic relocalization within PM nanodomains to promote functional signaling platforms, concomitant with modifications of PM order and fluidity, have been evidenced in immune signaling, host-pathogen interactions, and particularly documented in plant-microorganism interactions [7–10] (for a recent review see Jaillais and Ott, 2020 [10]). Furthermore, the asymmetry of the lipid distribution between the two leaflets of animal and plant PM is another key feature of membrane organization and function. In animals, most of the available data on asymmetry comes from red blood cells and is still not yet fully elucidated. In plants, very few publications partially examine these crucial questions. Work performed on oat root PM indicated that phospholipids dominate the cytosolic leaflet followed by total sterols, whereas the reverse order applies to the apoplastic leaflet of the oat root PM [11]. Investigating the molecular basis of the electrostatic characteristics of plant endomembranes, Jaillais and coll. evidenced that PA and PS sensors accumulate at the PM cytosolic leaflet in Arabidopsis root epidermis, together with PI 4-phosphate (PI4P) [12]. Recent data suggested that GIPCs might be mainly located in the outer leaflet of tobacco PM [6] but currently no indication about the localization of the different molecular species of free sterols nor lipid-associated fatty acids are available in the literature.
Additionally, the assertion from line 269 was wrong and was deleted.
2b) Later in the text, the authors often mention different phases and phase transitions; it is not explained at all how these pertain to the plant plasma membrane.
Response:
Indeed, the link between phase transition and biological function was implicit and not clear. To overcome this issue, we added the following paragraph when the gel/fluid phase transition is introduced:
Likewise, plotting M1 against the temperature is useful to analyze the changes in the lipid dynamics along with the temperature and to determine the phase transition temperature, Tm, where the lipid chains undergo a transition from almost static (gel phase) to highly mobile and disordered (fluid phase) (Figure 4). Such transition hardly occurs in vivo where biological functions require a well-balanced amount of lipid mobility to occur. Because a molecule that alters Tm has a direct impact on the lipid dynamics at a given temperature, it may enhance or reduce any biological functions that depend on it (e. g. signal transduction).
3) Maybe the term ‘perception’ can be used when describing an interaction with a receptor in the PM; since the authors discuss interactions with the lipid component of the PM, I would strongly recommend more technical terms ‘interaction’ or ‘binding’.
Response:
As already explained, 'perception' makes sense in a biological context when a molecule induces a biological response for instance. We corrected all the occurrences where 'perception' was used to describe actual molecular interactions. For example, section 4 'Plant plasma membrane perception of amphiphilic elicitor compounds' was renamed 'Interaction of amphiphilic elicitors with the plant plasma membrane'. All the remaining occurrences of 'perception' implying a biological context are perfectly suited.
4) Chapter on biomimetic membrane models: the authors do not comment at all on the lipid composition of model membranes. Only in line 160-161 they mention model membranes made solely of sterols; I fail to see how such a model could be informative about the processes occurring at the plant PM.
This his highly relevant for the stated goal of this review, which discusses contributions and limitations of biophysical approaches.
Response:
The corresponding section was updated and expanded. (See answer to remark 2a)
5) In general, figure legends are insufficient, figures are sloppily prepared and labelled. The use of fonts and size is inconsistent even within the same figure, let alone between figures
Response:
All the figures were fixed, altered, or remade from scratch to be consistent and more easily readable. All the captions were updated accordingly.
Figure 1: Please add a more specific title and some description. What is SUV/LUV/GUV/MLV, what does Ø refer to? Are the structures drawn to scale in panel A? What is the range of sizes for GUVs? In line 137, µM should be changed to µm. Panels are labelled A,B and C while in text they are i, ii, and iii.
Response:
Figure 1 was fixed and its caption was completed to be "self-sufficient". Vesicles from panel A are indeed drawn to scale (as much as size range and visibility allow); this note was added to the caption. GUVs and MLVs have the same size, this is why the diameter for those vesicles is written between the two depictions.
The corresponding sections in the main text were fixed and updated to be more coherent with Figure 1.
Figure 2: Can there also be a change in the polar head dynamics?
Response:
Figure 2 was updated to show all cases including the impact of an inserted molecule on polar head dynamics and the resulting 31P NMR spectrum.
Figure 3: Please define DMPC. There are many other labels in the figure that are not explained or too small to read. ‘Depaking’ is probably ‘Depeaking’?
Response:
Figure 3 was updated and simplified. the "depaking" (or "de-Pake-ing" from Pake doublet) procedure was removed from the figure as it is not mandatory. So mentioning it led to an unnecessarily complex figure.
In legend you can say ΔνQ is directly proportional to Scd
Response:
This now appears both in the legend and in the figure itself.
Figure 4: Title? Label “A” should be moved. In the text, please comment on the relevance of these phase transitions for the plant PM.
Response:
The whole figure was remade from scratch and the relevance of the gel/phase transition was added to the main text (see the response to remark 2b).
Figure 5: Title: Molecular modelling approaches. Inconsistent labelling, use A, B and C.
Use the same molecular representation in panels A and B. This figure is particularly sloppy and uninformative. In panels A/B, I don’t understand what is docked on what. The lower panel doesn’t explain much about molecular dynamics simulations. The equations are not explained at all, so I am not sure how they will be useful to a reader. A specific example, describing what information was obtained from the simulation, would be much more useful. I suppose that what is shown is an actual simulation, so why not explain it?
Response:
The whole figure was remade from scratch to be clearer.
Figure 6: More explanation is needed. Panel A is not clear; what is the molecule above the arrow? What are the black and yellow symbols representing? What is plotted in panel B? What are the probes shown in panel C, what is the difference between the left and the right image?
Response:
The figure was modified to be more readable and the legend was expanded.
6) Abbreviations throughout the text: some are defined twice, some are used only once (therefore not needed), many are not defined. This will be particularly confusing for a non-expert reader.
Response:
We paid attention to this issue by removing all unnecessary abbreviations and double-checking that they were properly defined before using them.
Minor comments:
313 I don’t know what the authors mean by “membrane activity” -> we removed the end of this sentence which was not clear nor really useful.
326 Period after “lipid” is missing -> It was added
395 “a few molar percent” clarify range of concentration. -> typically range was added
Some sentences were difficult to understand.
254 …to study the influence of elicitors on the temperature of the gel-to-fluid phase transition as it is sensible to lipid dynamics. -> changed to "the impact of elicitors on the temperature of the gel-to-fluid phase transition."
326 For instance, the study of the microbial cytolysin NLP has notably shown that plant specific glycolipids, GIPCs are involved in the recognition of the protein (should be referenced to Lenarcic 2017) -> the phrase was modified and the proper citation was used
440 Similar experiments on surfactin showed an inhibition of solubilization for liposomes containing PE but an enhancement of calcein release in presence of PG [101]. -> modified to "Similar experiments on surfactin showed it promotes calcein release when PG is present but inhibits the solubilization of liposomes that contain PE"
489/490 Bluishes/greenishes are not English words -> replaced by the proper verbs ("blues"/"greens")
In general, the manuscript requires editing for English language and clarity. -> We did our best to proofread the manuscript and correct broken syntax or grammar.
Round 2
Reviewer 2 Report
The authors have adequately addressed my comments and the manuscript has been significantly improved. All figures and figured legends have revised or remade, substantially improving the clarity of the review. The manuscript can be accepted for publication.